# INFORMATION LAUNDERING FOR MODEL PRIVACY

**Xinran Wang**
School of Statistics
University of Minnesota-Twin Cities
Minneapolis, MN 55455, USA
`wang8740@umn.edu`

**Yu Xiang**
Electrical and Computer Engineering
University of Utah
Salt Lake City, UT 84112, USA
`yu.xiang@utah.edu`

**Jun Gao**
Department of Mathematics
Stanford University
Stanford, CA 94305, USA
`jung2@stanford.edu`

**Jie Ding**
School of Statistics
University of Minnesota-Twin Cities
Minneapolis, MN 55455, USA
`dingj@umn.edu`

## ABSTRACT

In this work, we propose information laundering, a novel framework for enhancing model privacy. Unlike data privacy that concerns the protection of raw data information, model privacy aims to protect an already-learned model that is to be deployed for public use. The private model can be obtained from general learning methods, and its deployment means that it will return a deterministic or random response for a given input query. An information-laundered model consists of probabilistic components that deliberately maneuver the intended input and output for queries of the model, so the model's adversarial acquisition is less likely. Under the proposed framework, we develop an information-theoretic principle to quantify the fundamental tradeoffs between model utility and privacy leakage, and derive the optimal design.

## 1 INTRODUCTION

An emerging number of applications involve the following user-scenario. Alice developed a model that takes a specific query as input and calculates a response as output. The model is a stochastic black-box that may represent a novel type of ensemble models, a known deep neural network architecture with sophisticated parameter tuning, or a physical law described by stochastic differential equations. Bob is a user that sends a query to Alice and obtains the corresponding response for his specific purposes, whether benign or adversarial. Examples of the above scenario include many recent Machine-Learning-as-a-Service (MLaaS) services (Alabbadi, 2011; Ribeiro et al., 2015; Xian et al., 2020) and artificial intelligence chips, where Alice represents a learning service provider, and Bob represents users.

Suppose that Bob obtains sufficient paired input-output data as generated from Alice's black-box model, it is conceivable that Bob could treat it as supervised data and reconstruct Alice's model to some extent. From the view of Alice, her model may be treated as valuable and private. As Bob who queries the model may be benign or adversarial, Alice may intend to offer limited utility for the return of enhanced privacy. The above concern naturally motivates the following problem.

*(Q1) How to enhance the privacy for an already-learned model?* Note that the above problem is not about data privacy, where the typical goal is to prevent adversarial inference of the data information during data transmission or model training. In contrast, model privacy concerns an already-established model. We propose to study a general approach to jointly maneuver the original query's input and output so that Bob finds it challenging to guess Alice's core model. As illustrated in Figure 1a, Alice's model is treated as a transition kernel (or communication channel) that produces $\tilde{Y}$ conditional on any given $\tilde{X}$. Compared with an honest service Alice would have provided (Figure 1b), the input $\tilde{X}$ is a maneuvered version of Bob's original input $X$; Moreover, Alice may choose

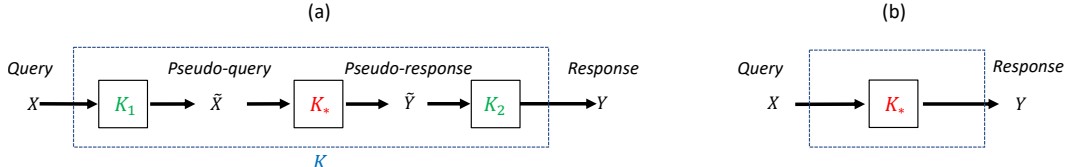

Figure 1: Illustration of (a) Alice's effective system for public use, and (b) Alice's idealistic system not for public use. In the figure, $K_*$ denotes the already-learned model/API, $K_1$ denotes the kernel that perturbs the input data query by potential adversaries, and $K_2$ denotes the kernel that perturbs the output response from $K_*$ to publish the final response $Y$.

to return a perturbed outcome $Y$ instead of $\tilde{Y}$ to Bob. Consequently, the apparent kernel from Bob's input query $X$ to the output response $Y$ is a cascade of three kernels, denoted by $K$ in Figure 1a. The above perspective provides a natural and general framework to study model privacy. Admittedly, if Alice produces a (nearly) random response, adversaries will find it difficult to steal the model, while benign users will find it useless. Consequently, we raise another problem.

*(Q2) How to formulate the model privacy-utility tradeoff, and what is the optimal way of imposing privacy?* To address this question, we formulate a model privacy framework from an information-theoretic perspective, named information laundering. We briefly describe the idea below. The general goal is to jointly design the input and output kernels ($K_1$ and $K_2$ in Figure 1a) that deliberately maneuver the intended input and output for queries of the model so that 1) the effective kernel ($K$ in Figure 1a) for Bob is not too far away from the original kernel ($K_*$ in Figure 1a), and 2) adversarial acquisition of the model becomes difficult. Alternatively, Alice 'launders' the input-output information maximally given a fixed utility loss. To find the optimal way of information laundering, we propose an objective function that involves two components: the first being the information shared between $X, \tilde{X}$ and between $\tilde{Y}, Y$, and the second being the average Kullback-Leibler (KL) divergence between the conditional distribution describing $K$ and $K_*$. Intuitively, the first component controls the difficulty of guessing $K_*$ sandwiched between two artificial kernels $K_1$ and $K_2$, while the second component ensures that overall utility is maximized under the same privacy constraints. By optimizing the objective for varying weights between the components, we can quantify the fundamental tradeoffs between model utility and privacy.

## 1.1 RELATED WORK

We introduce some closely related literature below. Section 3.3 will incorporate more technical discussions on some related but different frameworks, including information bottleneck, local data privacy, information privacy, and adversarial model attack.

A closely related subject of study is data privacy, which has received extensive attention in recent years due to societal concerns (Voigt & Von dem Bussche, 2017; Evans et al., 2015; Cross & Cavallaro, 2020; Google, 2019; Facebook, 2020). Data privacy concerns the protection of (usually personal) data information from different perspectives, including lossless cryptography (Yao, 1982; Chaum et al., 1988), randomized data collection (Evfimievski et al., 2003; Kasiviswanathan et al., 2011; Ding & Ding, 2020), statistical database query (Dwork & Nissim, 2004; Dwork, 2011), membership inference (Shokri et al., 2017), and Federated learning (Shokri & Shmatikov, 2015; Konevcny et al., 2016; McMahan et al., 2017; Yang et al., 2019; Diao et al., 2020). A common goal in data privacy is to obfuscate individual-level data values while still enabling population-wide learning. In contrast, the subject of model privacy focuses on protecting a single learned model ready to deploy. For example, we want to privatize a classifier to deploy on the cloud for public use, whether the model is previously trained from raw image data or a data-private procedure.

Another closely related subject is the model extraction in (Tramèr et al., 2016; Papernot et al., 2016b), where Bob's goal is to reconstruct Alice's model from several queries' inputs and outputs, knowing what specific model Alice uses. For example, suppose that Alice's model is a generalized linear regression with $p$ features. In that case, it is likely to be reconstructed using $p$ queries of the expected mean (a known function of $X\beta$) by solving equations (Tramèr et al., 2016). In the supervised classification scenario, when only labels are returned to any given input, model extraction could be

cast as an active learning problem where the goal is to query most efficiently (Chandrasekaran et al., 2018). Model extraction was also studied in contexts beyond the prediction API, e.g., when an adversary utilizes the gradient information (Milli et al., 2019). From Alice's perspective, there exist several solutions to safeguard against model leakage. A warning-based method was developed in (Kesarwani et al., 2018), where Alice continuously monitors the information gain and raise alarms when they become unusual. Another warning method was developed in (Juuti et al., 2019), where the detection of an adversary is based on testing whether the pairwise distances among queried data approximately follow the Gaussian distribution. The work in (Lee et al., 2018) developed a defense strategy for the setting where the target is a classifier, and the adversary queries each class's probability. The probabilities are maximally perturbed under the constraint that the most-likely class label) remains the same. The work in (Orekondy et al., 2019) studied a similar setting but from a different perspective. The main idea is to perturb the probabilities within an $\ell_2$-distance constraint to poison the adversary's gradient signals.

## 1.2 CONTRIBUTIONS AND OUTLINE

The main contributions of this work are three folds. First, we develop a novel concept, theory, and method, generally referred to as *information laundering*, to study model privacy. Unlike data privacy that concerns the protection of raw data information, model privacy aims to privatize an already-learned model for public use. To the best of the authors' knowledge, this work is the first framework to study model privacy in a principled manner. Second, under the developed information-theoretic framework, we cast the tradeoffs between model privacy and utility as a general optimization problem. We derive the optimal solution using the calculus of variations and provide extensive discussions on the solution's insights from different angles. Third, we develop a concrete algorithm, prove its convergence, and elaborate on some specific cases. We also provide some experimental studies to illustrate the concepts, and discuss several future problems at the end.

The paper is organized as follows. In Section 2, we describe the problem formulation and a general approach to protect the model. In Section 3, we propose the information laundering method that casts the model privacy-utility tradeoff as an optimization problem and derives a general solution. In Section 3.3, we provide some additional discussions of the related frameworks, including information bottleneck, local data privacy, information privacy, and adversarial model attack. In Section 5, we conclude the paper with some potential future work. In the Appendix, we provide the proofs of the main results and experimental studies.

## 2 FORMULATION

### 2.1 BACKGROUND

The private model can be obtained from general learning methods, and its deployment means that it will return a response for a given input query. Suppose that $\mathcal{X}$ and $\mathcal{Y}$ are the input and output alphabets (data space), respectively.

**Definition 1 (Learned model)** *A learned model is a kernel $p : \mathcal{X} \times \mathcal{Y} \to [0, 1]$, which induces a class of conditional distributions $\{p(\cdot \mid x) : x \in \mathcal{X}\}$.*

A model in the above definition is also referred to as a communication channel in information theory. A model can be regarded as the input-output (or Alice's application programming interface, API) offered to Bob. Examples include a regression/classification model that outputs predicted labels, a clustering model that outputs the probabilities of belonging to specific groups, and a stochastic differential equation system that outputs the likely paths for various inputs variables. It does not matter where the model comes from since we are only concerned about the privacy of a fixed given model. The (authentic) model of Alice is denoted by $p_{K_*}$.

An adversary Bob is a user that can access the above model's API, providing an arbitrary input, $X$, and obtaining an output, $Y$. Bob aims to use as few queries as possible to construct a model that closely matches Alice's model $p_{K_*}$. We will formalize the 'closeness' using the KL divergence.

What is model privacy? Our perspective is that privacy is not an intrinsic quantity associated with a model; instead, it is a measure of information that arises from interactions between the model and

its queries. In our context, the interactions are through $X$ (offered by Bob) and $Y$ (offered by Alice). The key idea of enhancing Alice's model privacy is to let Alice output noisy predictions $\tilde{Y}$ for any input $X$ so that Bob cannot easily infer Alice's original model. Similarly, Alice may choose to manipulate $X$ as well before passing it through $K_*$. Alternatively, Alice intends to 1) impose some ambiguity between $X, \tilde{X}$, and between $Y, \tilde{Y}$, which conceivably will produce response deviating from the original one, and 2) seek the $K$ closest to $K_*$ under the same amount of ambiguity imposed. Motivated by the above concepts, we introduce the following notion.

**Definition 2 (Information-laundered model)** *A information-laundered model with respect to a given model $K_*$ is a model $K$ that consists of three internal kernels $K = K_1 \circ K_* \circ K_2$ (illustrated in Figure 1).*

Naturally, the information-laundered model of Alice is denoted by $p_K$.

## 2.2 NOTATION

We let $p_{K_*}(\cdot \mid \cdot), p_{K_1}(\cdot \mid \cdot), p_{K_2}(\cdot \mid \cdot), p_K(\cdot \mid \cdot)$ denote the kernels that represent the authentic model, input kernel, output kernel, and the information-laundered model, respectively. We let $p_X(\cdot)$ denote the marginal distribution of $X$. Similar notation is for $p_{\tilde{X}}(\cdot), p_{\tilde{Y}}(\cdot)$, and $p_Y(\cdot)$. Note that the $p_Y$ implicitly depends on the above conditional distributions. We use $p_{K_1 \circ K_*}(\cdot \mid \cdot)$ and $p_{K_* \circ K_2}(\cdot \mid \cdot)$ to denote cascade conditional distributions of $\tilde{Y} \mid X$ and $Y \mid \tilde{X}$, respectively.

Throughout the paper, random variables are denoted by capital letters. Suppose that $X \in \mathcal{X}, \tilde{X} \in \tilde{\mathcal{X}}$, $\tilde{Y} \in \tilde{\mathcal{Y}}$, and $Y \in \mathcal{Y}$. For technical convenience, we will assume that $\mathcal{X}, \tilde{\mathcal{X}}, \tilde{\mathcal{Y}}, \mathcal{Y}$ are finite alphabets unless otherwise stated. We will discuss some special cases when some of them are the same. Our theoretical results apply to continuous alphabets as well under suitable conditions. For notational convenience, we write the sum $\sum_{x \in \mathcal{X}} u(x)$ as $\sum_x u(x)$ for any function $u$.

With a slight abuse of notation, we will use $p$ to denote a distribution, density function, or transition kernel, depending on the context.

## 3 INFORMATION LAUNDERING

### 3.1 THE INFORMATION LAUNDERING PRINCIPLE

The information laundering method is an optimization problem formulated from the concept of KL-divergence between the (designed) effective kernel and the original kernel, with constraints of the privacy leakage during the model-data interaction. In particular, we propose to minimize the following objective function over $(p_{K_1}, p_{K_2})$,

$$L(p_{K_1}, p_{K_2}) \triangleq \mathbb{E}_{X \sim p_X} D_{\mathrm{KL}}(p_{K_*}(\cdot \mid X), p_K(\cdot \mid X)) + \beta_1 I(X; \tilde{X}) + \beta_2 I(Y; \tilde{Y}). \tag{1}$$

In the above, $K_1$ and $K_2$ are implicitly involved in each additive term of $L$, and $\beta_1 \geq 0, \beta_2 \geq 0$ are constants that determine the utility-privacy tradeoffs. Small values of $\beta_1$ and $\beta_2$ (e.g., zeros) pushes the $K$ to be the same as $K_*$, while large values of $\beta_1$ pushes $\tilde{X}$ to be nearly-independent with $X$ (similarly for $\beta_2$). It is worth mentioning that the principle presumes a given alphabet (or representation) for $\tilde{X}$ and $\tilde{Y}$. The variables to optimize over is the transition laws $X \to \tilde{X}$ and $\tilde{Y} \to Y$.

The objective in (1) may be interpreted in the following way. On the one hand, Alice aims to develop an effective system of $K$ that resembles the authentic one $K_*$ for the utility of benign users. This goal is realized through the first term in (1), which is the average divergence between two system dynamics. On the other hand, Alice's model privacy leakage is through interactions with Bob, which in turn is through the input $X$ (publicly offered by Bob) and output $Y$ (publicly offered by Alice). Thus, we control the information propagated through both the input-interfaces and out-interfaces, leading to the second and third terms in (1).

We note that the above objective function may also be formulated in alternative ways from different perspectives. For example, we may change the third term to be $\beta_2 I(Y; \tilde{Y} \mid X, \tilde{X})$, interpreted

in the way that Alice will design $K_1$ first, and then design $K_2$ conditional on $K_1$. Likewise, we may change the second term to be $\beta_1 I(X; \tilde{X} \mid \tilde{Y}, Y)$, meaning that $K_2$ is designed first. From Bob's perspective, we may also change the third term to $\beta_2 I(Y; \tilde{Y} \mid X)$, interpreted for the scenario where Bob conditions on the input information $X$ during model extraction. Additionally, from the perspective of adaptive interactions between Alice and Bob, we may consider $p_X$ as part of the optimization and solve the max-min problem $\max_{p_X} \min_{p_{K_1}, p_{K_2}} L(p_{K_1}, p_{K_2})$. We leave these alternative views for future work.

## 3.2 THE OPTIMAL SOLUTION

We derive the solution that corresponds to the optimal tradeoffs and point out some nice interpretations of the results. The derivation is nontrivial as the functional involves several nonlinear terms of the variables to optimize over. Note that for the notation defined in Subsection 2.2, only $p_X$ and $p_{K_*}$ are known and others are (implicitly) determined by $p_{K_1}, p_{K_2}$.

**Theorem 1** *The optimal solution of (1) satisfies the following equations.*

$$p_{K_1}(\tilde{x} \mid x) = \kappa_x p_{\tilde{X}}(\tilde{x}) \exp\left\{ \frac{1}{\beta_1} \mathbb{E}_{Y|X=x \sim p_{K_*}} \frac{p_{K_* \circ K_2}(Y \mid \tilde{x})}{p_K(Y \mid x)} - \frac{\beta_2}{\beta_1} \mathbb{E}_{\tilde{Y}, Y|\tilde{X}=\tilde{x}} \log \frac{p_{K_2}(Y \mid \tilde{Y})}{p_Y(Y)} \right\},$$
(2)

$$p_{K_2}(y \mid \tilde{y}) = \tau_{\tilde{y}} p_Y(y) \exp\left\{ \frac{1}{\beta_2 p_{\tilde{Y}}(\tilde{y})} \mathbb{E}_{X \sim p_X} \frac{p_{K_*}(y \mid X) \cdot p_{K_1 \circ K_*}(\tilde{y} \mid X)}{p_K(y \mid X)} \right\},$$
(3)

*where $\kappa_x$ and $\tau_{\tilde{y}}$ are normalizing constants implicitly defined so that the conditional density function integrates to one.*

Note that the distributions of $\tilde{X}, \tilde{Y}, Y$, and $\tilde{Y}, Y \mid \tilde{X}$, implicitly depend on $p_{K_1}$ and $p_{K_2}$. The above theorem naturally leads to an iterative algorithm to estimate the unknown conditional distributions $p_{K_1}$ and $p_{K_2}$. In particular, we may alternate Equations (2) and (3) to obtain $p_{K_1}^{(\ell)}(\tilde{x} \mid x), p_{K_2}^{(\ell)}(y \mid \tilde{y})$ from $p_{K_1}^{(\ell-1)}(\tilde{x} \mid x), p_{K_2}^{(\ell-1)}(y \mid \tilde{y})$ at step $\ell = 1, 2, \ldots$ with random initial values at $\ell = 0$. The pseudocode is summarized in Algorithm 1.

In the next theorem, we show that the convergence of the algorithm. The sketch of the proof is described below. First, we treat the original objective $L$ as another functional $J$ of four independent variables, $p_{K_1}, p_{K_2}, h_1, h_2$, evaluated at $h_1 = p_{\tilde{X}}$ and $h_2 = p_Y$. Using a technique historically used to prove the convergence of the Blahut-Arimoto algorithm for calculating rate-distortion functions in information theory, we show that $J \geq L$. We also show that $L$ is convex in each variable so that the objective function is non-increasing in each alternation between four equations. Since $L \geq 0$, the convergence is implied by the monotone convergence theorem.

**Theorem 2** *Algorithm 1 converges to a minimum that satisfies equations (2) and (3).*

Note that the minimum is possibly a local minimum. We will later show the convergence to a global minimum in a particular case. Next, we provide interpretations of the parameters and how they affect the final solution.

A large $\beta_1$ in the optimization of (1) indicates a higher weight on the term $I(X; \tilde{X})$. In the extreme case when $\beta_1 = \infty$, minimizing $I(X; \tilde{X})$ is attained when $\tilde{X}$ is independent with $X$. Consequently, the effective model of Alice produces a fixed distribution of responses for whatever Bob queries. The above observation is in line with the derived equation (2), which will become $p_{K_1}(\tilde{x} \mid x) \approx \kappa_x p_{\tilde{X}}(\tilde{x})$ (and thus $\kappa_x \approx 1$) for a large $\beta_1 > 0$.

Similar to the effect of $\beta_1$, a larger $\beta_2$ imposes more independence between $\tilde{Y}$ and $Y$. In the case $\beta_2 = \infty$, Alice may pass the input to her internal model $K_*$ but output random results. This can be seen from either the Formulation (1) or Equation (3).

For the first expectation in equation (2), the term may be interpreted as the average likelihood ratio of $y$ conditional on $\tilde{x}$ against $x$. From Equation (2), it is more likely to transit from $x$ to $\tilde{x}$ in the presence of a larger likelihood ratio. This result is intuitively appealing because a large likelihood ratio indicates that $x$ may be replaced with $\tilde{x}$ without harming the overall likelihood of observing $Y$.

---

**Algorithm 1** Optimized Information Laundering (OIL)

---

**input** Input distribution $p_X$, private model $p_{K_*}$, alphabets $\mathcal{X}, \tilde{\mathcal{X}}, \tilde{\mathcal{Y}}, \mathcal{Y}$ for $X, \tilde{X}, \tilde{Y}, Y$, respectively.
**output** Transition kernels $p_{K_1}$ and $p_{K_2}$

1: Let $p_{\tilde{X}}^{(0)}$ and $p_Y^{(0)}$ denote the uniform distribution on $\tilde{\mathcal{X}}$ and $\mathcal{Y}$, respectively.
2: **for** $t = 0 \to T - 1$ **do**
3:  Calculate

$$p_{K_1}^{(t+1)}(\tilde{x} \mid x) = \kappa_x p_{\tilde{X}}^{(t)}(\tilde{x}) \exp\left\{ \frac{1}{\beta_1} \mathbb{E}_{Y \mid x \sim p_{K_*}} \frac{p_{K_* \circ K_2}^{(t)}(Y \mid \tilde{x})}{p_K^{(t)}(Y \mid x)} \right.$$

$$\left. - \frac{\beta_2}{\beta_1} \mathbb{E}_{\tilde{Y}, Y \mid \tilde{x} \sim p_{K_* \circ K_2}^{(t)}} \log \frac{p_{K_2}^{(t)}(Y \mid \tilde{Y})}{p_Y^{(t)}(Y)} \right\},$$

$$p_{K_2}^{(t+1)}(y \mid \tilde{y}) = \tau_{\tilde{y}} p_Y^{(t)}(y) \exp\left\{ \frac{1}{\beta_2 p_{\tilde{Y}}^{(t)}(\tilde{y})} \mathbb{E}_{X \sim p_X} \frac{p_{K_*}(y \mid X) \cdot p_{K_1 \circ K_*}^{(t+1)}(\tilde{y} \mid X)}{p_K^{(t+1,t)}(y \mid X)} \right\},$$

$$p_{\tilde{X}}^{(t+1)}(\tilde{x}) = \sum_x p_{K_1}^{(t+1)}(\tilde{x} \mid x) p_X(x),$$

$$p_Y^{(t+1)}(y) = \sum_{\tilde{y}} p_{K_2}^{(t+1)}(y \mid \tilde{y}) p_{\tilde{Y}}^{(t+1)}(\tilde{y}),$$

where $p_{K_1 \circ K_*}^{(t+1)}, p_{K_* \circ K_2}^{(t)}$, and $p_K^{(t+1,t)}$ denote the kernels cascaded from $(p_{K_1}^{(t+1)}, p_{K_*})$, $(p_{K_*}, p_{K_2}^{(t)})$, and $(p_{K_1}^{(t+1)}, p_{K_*}, p_{K_2}^{(t)})$, respectively, and $p_{\tilde{Y}}^{(t+1)}$ is the marginal from $(p_{\tilde{X}}^{(t+1)}, p_{K_*}, p_{K_2}^{(t+1)})$.
4: **end for**
5: Return $p_{K_1} = p_{K_1}^{(T)}, p_{K_2} = p_{K_2}^{(T)}$.

---

## 3.3 FURTHER DISCUSSIONS ON RELATED WORK

**Information Bottleneck: extracting instead of privatizing information**. The information bottleneck method (Tishby et al., 2000) is an information-theoretic approach that aims to find a parsimonious representation of raw data $X$, denoted by $\tilde{X}$, that contains the maximal information of a variable $Y$ of interest. The method has been applied to various learning problems such as clustering, dimension reduction, and theoretical interpretations for deep neural networks (Tishby & Zaslavsky, 2015). Formally, the information bottleneck method assumes the Markov chain

$$\tilde{X} \to X \to Y, \tag{4}$$

and seeks the the optimal transition law from $X$ to $\tilde{X}$ by minimizing the functional

$$L(p_{\tilde{X} \mid X}) = I(X; \tilde{X}) - \beta I(\tilde{X}; Y),$$

with $\beta$ being is a tuning parameter that controls the tradeoffs between compression rate (the first term) and amount of meaningful information (second term). The alphabet of the above $\tilde{X}$ needs to be pre-selected and often much smaller in size compared to the alphabet of $X$ to meet the purpose of compression. In other words, the information that $X$ provides about $Y$ is passed through a 'bottleneck' formed by the parsimonious alphabet of $\tilde{X}$.

A similarity between the information bottleneck method and the particular case of information laundering in Subsection A.2 is that they both optimize a functional of the transition law of $X \to \tilde{X}$. Nevertheless, their objective and formulation are fundamentally different. First, the objective of information bottleneck is to compress the representation while preserving meaningful information, under the assumption of (4); Our goal is to distort $X$ while minimizing the gap between the (random) functionality of $X \to Y$, under a different Markov chain $X \to \tilde{X} \to Y$.

**Data Privacy and Information Privacy: protecting data instead of a model**. The tradeoffs between individual-level data privacy and population-level learning utility have motivated active research on what is generally referred to as 'local data privacy' across multiple fields such as data mining (Evfimievski et al., 2003), security (Kasiviswanathan et al., 2011), statistics (Duchi et al., 2018), and information theory (du Pin Calmon & Fawaz, 2012; Sun et al., 2016). For example, a popular framework is the local differential privacy (Evfimievski et al., 2003; Dwork et al., 2006;

Kasiviswanathan et al., 2011), where raw data $X$ is suitably randomized (often by adding Laplace noises) into $Y$ so that the ratio of conditional densities

$$e^{-\alpha} \leq \frac{p_{Y|X}(y \mid x_1)}{p_{Y|X}(y \mid x_2)} \leq e^{\alpha} \tag{5}$$

for any $y, x_1, x_2 \in \mathcal{X}$, where $\alpha > 0$ is a pre-determined value that quantities the level of privacy. In the above, $X$ and $Y$ represent the private data and the processed data to be collected or publicly distributed. The requirement (5) guarantees that the KL-divergence between $p_{Y|x_1}$ and $p_{Y|x_2}$ is universally upper-bounded by a known function of $\alpha$ (see, e.g., Duchi et al., 2018), meaning that $x_1$ and $x_2$ are barely distinguishable from the observed $y$. Note that the above comparison is made between two conditional distributions, while the comparison in information laundering (recall the first term in (1)) is made between two transition kernels.

The local differential privacy framework does not need to specify a probability space for $X$, since the notion of data privacy is only built on conditional distributions. Another related framework is the information privacy (du Pin Calmon & Fawaz, 2012), which assumes a probabilistic structure on $X$ and a Markov chain $X \rightarrow \tilde{Y} \rightarrow Y$. In the above chain, $X$ is the private raw data, $\tilde{Y}$ is a set of measurement points to transmit or publicize, and $Y$ is a distortion of $\tilde{Y}$ that is eventually collected or publicized. We deliberately chose the above notation of $X, \tilde{Y}, Y$, so that the Markov chain appears similar to the special case of information laundering in Subsection 4. Nevertheless, the objective of information privacy is to minimize $I(X; Y)$ over $p_{Y|\tilde{Y}}$ subject to utility constraints, assuming that the joint distribution of $X, \tilde{Y}$ is known. In other words, the goal is to maximally hide the information of $X$. In the context of information laundering, the system input $X$ is provided by users and is known.

**Adversarial Model Attack: rendering harm instead of utility to a model**. The adversarial model attack literature concerns the adversarial use of specially crafted input data to cause a machine learning model, often a deep neural network, to malfunction (Papernot et al., 2016a; Narodytska & Kasiviswanathan, 2017; Papernot et al., 2017). For example, an adversarial may inject noise into an image so that a well-trained classifier produces an unexpected output, even if the noise is perceptually close to the original one. A standard attack is the so-called (Adaptive) Black-Box Attack against classifiers hosted by a model owner, e.g., Amazon and Google (Rosenberg et al., 2017; Chakraborty et al., 2018). For a target model $K_*$, a black-box adversary has no information about the training process of $K_*$ but can access the target model through query-response interfaces. The adversary issues (adaptive) queries and record the returned labels to train a local surrogate model. The surrogate model is then used to craft adversarial samples to maximize the target model's prediction error.

If we let $X, \tilde{X}, Y$ denote the model input, adversarially perturbed input, and output, respectively, then we may draw a similarity between adversarial model attack and the particular case of information laundering in Subsection A.2 since they both look for the law $X \rightarrow \tilde{X}$. The main difference is in the objective. While the model attack aims to find an input domain that maximally distorts the model, information laundering aims to maintain a small model discrepancy. Under our notation, a possible formulation for the model attack is to seek $\max_{p_{\tilde{X}|X}} \mathbb{E}_{X \sim p_X} D_{\mathrm{KL}}(p_{K_*}(\cdot \mid X), p_{K_*}(\cdot \mid \tilde{X}))$ under a constraint on $p_{\tilde{X}|X}$.

## 4 SPECIAL CASE: INFORMATION LAUNDERING OF THE OUTPUT ($Y$) ONLY

Two special cases of an information-laundered system are illustrated in Figure 2. Here, we elaborate on one case and include the other special case in the Appendix. Suppose that $K_1$ is an identity map and let $\beta_1 = 0$. In other words, we alter the output data only (Figure 2b). Furthermore, suppose that for each given $\tilde{x}$, the conditional distribution $p_K(\cdot \mid \tilde{x})$ assigns all the mass at $\tilde{y}$. In other words, $K_*$ reduces to a deterministic function mapping from each $\tilde{x} \in \mathcal{X}$ to a unique $\tilde{y} \in \mathcal{Y}$, which is denoted by $\tilde{y} = f(\tilde{x})$. For example, Alice's model is a classifier that takes input features and returns hard-thresholded classification labels. Then the optimization problem (1) reduces to minimizing

$$L(p_{K_2}) \triangleq \mathbb{E}_{X \sim p_X} D_{\mathrm{KL}}(p_{K_*}(\cdot \mid X), p_K(\cdot \mid X)) + \beta_2 I(Y; \tilde{Y}). \tag{6}$$

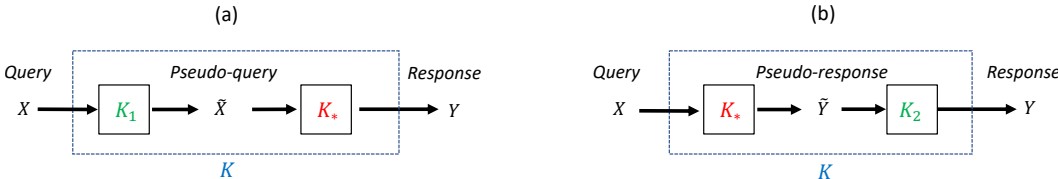

Figure 2: Illustration of Alice's information-laundered system for public use, by (a) alternating input only, and (b) alternating output only. The notations are similar to those in Figure 1.

**Corollary 1** *The solution to the optimization problem (6) satisfies*

$$p_{K_2}(y \mid \tilde{y}) = \tau_{\tilde{y}} p_Y(y) \exp\left\{ \frac{1}{\beta_2 p_{\tilde{Y}}(\tilde{y})} \mathbb{E}_{X \sim p_X} \frac{p_{K_*}(y \mid X) \cdot p_{K_*}(\tilde{y} \mid X)}{p_K(y \mid X)} \right\}, \qquad (7)$$

*where $\tau_{\tilde{y}}$ is a normalizing constant. In particular, if $K_*$ is deterministic, equation (7) becomes*

$$p_{K_2}(y \mid \tilde{y}) = \tau_{\tilde{y}} p_Y(y) \exp\left\{ \frac{1}{\beta_2 p_{\tilde{Y}}(\tilde{y})} \sum_{x: f(x)=y} p_X(x) \frac{\mathbb{1}_{y=\tilde{y}}}{p_K(y \mid x)} \right\}$$

$$= \tau_{\tilde{y}} p_Y(y) \exp\left\{ \frac{\mathbb{1}_{y=\tilde{y}}}{\beta_2 \, p_{K_2}(y \mid y)} \right\} \qquad (8)$$

To exemplify the proposed methodology, we study a specific case with the following conditions.
1) $\mathcal{X}$ may be large or continuously-valued, $\tilde{\mathcal{Y}} = \mathcal{Y}$ is a moderately-large alphabet,
2) $\tilde{\mathcal{Y}} = \mathcal{Y}$ so that $\tilde{Y}$ and $Y$ are in the same space,
3) $K_*$ is deterministic.

Under the above scenario, we can apply Algorithm 1 and Corollary 1 to obtain a simplified procedure below (denoted by OIL-Y). At each time step $t = 1, 2, \ldots,$, for each $\tilde{y}, y \in \mathcal{Y}$, we calculate

$$p_{K_2}^{(t+1)}(y \mid \tilde{y}) = \tau_{\tilde{y}} p_Y^{(t)}(y) \exp\left\{ \frac{\mathbb{1}_{y=\tilde{y}}}{\beta_2 \, p_{K_2}^{(t)}(y \mid y)} \right\}, \text{ where } \tau_{\tilde{y}}^{-1} = \sum_y p_Y^{(t)}(y) \exp\left\{ \frac{\mathbb{1}_{y=\tilde{y}}}{\beta_2 \, p_{K_2}^{(t)}(y \mid y)} \right\},$$

$$p_Y^{(t+1)}(y) = r_{\tilde{y}} p_{K_2}^{(t+1)}(y \mid \tilde{y}), \text{ where } r_{\tilde{y}} = \sum_{x: f(x)=\tilde{y}} p_X(x). \qquad (9)$$

Note that the above $r_{\tilde{y}}$ is the probability that Alice observes $\tilde{y}$ as an output of $K_* *$ if Bob inputs $X \in p_X$. Therefore, $r_{\tilde{y}}$ can be easily estimated to be the empirical frequency of observing $\tilde{y}$ at the end of Alice.

Note that since $\mathcal{Y}$ is a finite alphabet, we can use a matrix representation for easy implementation. In particular, we represent the elements of $\mathcal{Y}$ by $1, \ldots, a$, where $a = \text{card}(\mathcal{Y})$. We then represent $p_{K_2}$ by $\boldsymbol{P} \in \mathbb{R}^{a \times a}$, and $p_Y$ by $\boldsymbol{q} \in \mathbb{R}^a$, where $P_{y,\tilde{y}} = p_{K_2}(y \mid \tilde{y})$. Such a representation will lead to a matrix form of the above procedure, summarized in Algorithm 2.

---

**Algorithm 2** OIL-Y (a special case of Algorithm 1, in the matrix form)

---

**input** Input distribution $p_X$, private model $p_{K_*}$
**output** Transition kernels $p_{K_2} : \mathcal{Y} \times \mathcal{Y} \to [0, 1]$ represented by $\boldsymbol{P} \in \mathbb{R}^{a \times a}$, where $a = \text{card}(\mathcal{Y})$
 1: Estimate $\boldsymbol{r} = [r_1, \ldots, r_a]$ from $p_X$ and $p_{K_*}$ as in equation (9)
 2: Initialize the entries of $\boldsymbol{P}^{(0)}$ and $\boldsymbol{q}^{(0)}$ (respectively representing $p_{K_2}, p_Y$) to be $1/a$
 3: **for** $t = 0 \to T - 1$ **do**
 4:    Calculate $\boldsymbol{P}^{(t+1)} = \boldsymbol{q}^{(t)} \times \boldsymbol{1}^T, \text{diag}(\boldsymbol{P})$, where $\boldsymbol{1} = [1, \ldots, 1]$ denote the $a \times 1$ vector.
 5:    Update $\text{diag}(\boldsymbol{P}^{(t+1)}) \leftarrow \text{diag}(\boldsymbol{P}^{(t+1)}) \cdot \exp\{1/(\beta_2 \text{diag}(\boldsymbol{P}^{(t)}))\}$, where the operations are element-wise
 6:    Scale each column (conditional distribution) of $\boldsymbol{P}^{(t+1)}$ so that it sums to one
 7:    Calculate $\boldsymbol{q}^{(t+1)} = \boldsymbol{P}^{(t+1)} \times \boldsymbol{r}$
 8: **end for**
 9: Return $p_{K_2}^{(T)}$ that is represented by $\boldsymbol{P}^{(T)}$.

---

Moreover, we proved the convergence to the global minimum for the alternating equations in the above scenario. The same technique can be emulated to show a similar result when we employ $K_1$ (instead of $K_2$) only. The result is summarized in Theorem 3.

**Theorem 3** *Suppose that $K_*$ is deterministic. The alternating equation (9), or its matrix form in Algorithm 1, converges to a global minimum of the problem (6).*

## 5 CONCLUSION AND FURTHER REMARKS

Despite extensive studies on data privacy, little has been studied for enhancing model privacy. Motivated by the emerging concern of model privacy from the perspective of machine learning service providers, we develop a plug-and-play methodology "information laundering" to enhance the privacy of any given model of interest. The information laundering is model-agnostic as it applies to general API models, including classification and regression models that output labels/probabilities, and black-box models generating probabilistic outputs. We believe that the developed principles, theories, and insights can lead to new resilient machine learning algorithms and services.

An interesting problem is to integrate information laundering with various application scenarios on a case-by-case basis. Another problem is to adapt the developed principle to specific API models to incorporate side information (also mentioned in Section 3.1). Taking into account the potential side information available to an adversary can be essential in some pathetic situations. Consider an example where the adversary knows that the output associated with extremely large input is a fixed constant. Then, the adversary may strategically send the same input with extreme values to accurately identify that constant. An information-laundered model may be vulnerable in the above scenario since the current information laundering concerns the average over data distributions.

Theoretically, there are three open problems left from the work that deserves further research. First, how does the imposed constraint of mutual information affect the rate of convergence from the adversary perspective for specific models (e.g., generalized linear models, decision trees, neural networks)? Second, we focused on finite alphabets for technical convenience. How to emulate the current methods for continuously-valued alphabets (especially with large dimensions)? Third, what would be the relative importance of laundering $X$ versus $Y$, and will this depend on specific learning problems?

**Appendix**. In Appendices A.1 and A.2, we first include two particular cases of information laundering that were not included in the main part of the paper. We then include the proofs of the theorems in Appendix A.3. Experimental results are included in Appendices A.4, A.5, A.6, and A.7 to demonstrate the algorithm convergence, model privacy-utility tradeoffs, how tradeoff parameters and unbalanced samples may influence the optimized information laundering, and how the information laundering effectively mitigates adversarial attacks.

ACKNOWLEDGMENTS

The last author was supported by the Army Research Office (ARO) under grant number W911NF-20-1-0222.

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

# A APPENDIX

## A.1 SPECIAL CASES: DETERMINISTIC MODEL $K_*$

In this case, Theorem 1 implies the following corollary. We will use this result in later sections.

**Corollary 2** *The optimal solution of (1) satisfies the following equations.*

$$
p_{K_1}(\tilde{x} \mid x) = \kappa_x p_{\tilde{X}}(\tilde{x}) \exp\left\{ \frac{1}{\beta_1} \frac{p_{K_2}(f(x) \mid f(\tilde{x}))}{\sum_{\tilde{x}'} p_{K_2}(f(x) \mid f(\tilde{x}')) p_{K_1}(\tilde{x}' \mid x)} \right.
$$
$$
\left. - \frac{\beta_2}{\beta_1} \mathbb{E}_{Y \mid \tilde{Y} = f(\tilde{x})} \log \frac{p_{K_2}(Y \mid f(\tilde{x}))}{p_Y(Y)} \right\},
$$
$$
p_{K_2}(y \mid \tilde{y}) = \tau_{\tilde{y}} p_Y(y) \exp\left\{ \frac{1}{\beta_2 p_{\tilde{Y}}(\tilde{y})} \sum_{x: f(x) = y} p_X(x) \frac{p_{K_1 \circ K_*}(\tilde{y} \mid x)}{p_K(y \mid x)} \right\},
$$

*where $\kappa_x$ and $\tau_{\tilde{y}}$ are normalizing constants implicitly defined so that the conditional density function integrates to one.*

## A.2 INFORMATION LAUNDERING OF THE INPUT ($X$) ONLY

Suppose that $K_2$ is an identity map and let $\beta_2 = 0$ so that we only maneuver the input data (Figure 2a). Then the optimization problem (1) reduces to minimizing

$$
L(p_{K_1}) \triangleq \mathbb{E}_{X \sim p_X} D_{\text{KL}}(p_{K_*}(\cdot \mid X), p_K(\cdot \mid X)) + \beta_1 I(X; \tilde{X}). \tag{10}
$$

**Corollary 3** *The optimal solution of (10) satisfies the following equations.*

$$
p_{K_1}(\tilde{x} \mid x) = \kappa_x p_{\tilde{X}}(\tilde{x}) \exp\left\{ \frac{1}{\beta_1} \mathbb{E}_{Y \mid X = x \sim p_{K_*}} \frac{p_{K_*}(Y \mid \tilde{X} = \tilde{x})}{p_K(Y \mid X = x)} \right\}, \tag{11}
$$

*where $\kappa_x$ is an implicitly defined normalizing constant. In particular, if $K_*$ is deterministic, equation (11) becomes*

$$
p_{K_1}(\tilde{x} \mid x) = \kappa_x p_{\tilde{X}}(\tilde{x}) \exp\left\{ \frac{\mathbb{1}_{f(x) = f(\tilde{x})}}{\beta_1 \sum_{\tilde{x}': f(x) = f(\tilde{x}')} p_{K_1}(\tilde{x}' \mid x)} \right\}. \tag{12}
$$

As we can see from Corollaries 1 and 3, for a deterministic $K_*$ (represented by $f$), the simplified equation of (8) is similar to that of (12). The subtle difference that one has a sum while the other does not is because $f$ may not be a one-to-one mapping.

## A.3 PROOFS

**Proof 1 (Proof of Theorem 1)** *Introducing Lagrange multipliers, $\lambda_1(x)$ for the normalization of the conditional distributions $p_{K_1}(\cdot \mid x)$ at each $x$, $\lambda_2(\tilde{y})$ for the normalization of the conditional distributions $p_{K_2}(\cdot \mid \tilde{y})$ at each $\tilde{y}$. The Lagrangian of (1) can be written as*

$$
L = -\sum_{x,y} p_X(x) p_{K_*}(y \mid x) \log p_K(y \mid x) + \beta_1 \sum_{x,\tilde{x}} p_X(x) p_{K_1}(\tilde{x} \mid x) \log \frac{p_{K_1}(\tilde{x} \mid x)}{p_{\tilde{X}}(\tilde{x})}
$$
$$
+ \beta_2 \sum_{\tilde{y}, y} p_{\tilde{Y}}(\tilde{y}) p_{K_2}(y \mid \tilde{y}) \log \frac{p_{K_2}(y \mid \tilde{y})}{p_Y(y)} + \sum_x \lambda_1(x) p_{K_1}(\tilde{x} \mid x) + \sum_{\tilde{y}} \lambda_2(\tilde{y}) p_{K_2}(y \mid \tilde{y}) + c
$$
$$
= A_1 + A_2 + A_3 + A_4 + A_5 + c \tag{13}
$$

*up to an additive constant $c$ that is determined by the known $p_X$ and $p_{K_*}$.*

*It can be verified that*

$$\frac{\partial p_K(y \mid x)}{p_{K_1}(\tilde{x} \mid x)} = p_{K_* \circ K_2}(y \mid \tilde{x}) \tag{14}$$

$$\frac{\partial p_{\tilde{X}}(\tilde{x})}{p_{K_1}(\tilde{x} \mid x)} = p_X(x) \tag{15}$$

$$\frac{\partial p_{\tilde{Y}}(\tilde{y})}{p_{K_1}(\tilde{x} \mid x)} = p_X(x) p_{K_*}(\tilde{y} \mid \tilde{x}) \tag{16}$$

$$\frac{\partial p_Y(y)}{p_{K_1}(\tilde{x} \mid x)} = p_X(x) p_{K_* \circ K_2}(y \mid \tilde{x}). \tag{17}$$

*Using (14)-(17), for a given $x$ and $\tilde{x}$, we calculate the derivatives of each term in (13) with respect to $p_{K_1}(\tilde{x} \mid x)$ to be*

$$\frac{\partial A_1}{p_{K_1}(\tilde{x} \mid x)} = -p_X(x) \sum_y p_{K_*}(y \mid x) \frac{p_{K_* \circ K_2}(y \mid \tilde{x})}{p_K(y \mid x)} \tag{18}$$

$$\frac{\partial A_2}{p_{K_1}(\tilde{x} \mid x)} = \beta_1 p_X(x) \log \frac{p_{K_1}(\tilde{x} \mid x)}{p_{\tilde{X}}(\tilde{x})} \tag{19}$$

$$\frac{\partial A_3}{p_{K_1}(\tilde{x} \mid x)} = \beta_2 p_X(x) \sum_{\tilde{y}, y} p_{K_* \circ K_2}(\tilde{y}, y \mid \tilde{X} = \tilde{x}) \log \frac{p_{K_2}(y \mid \tilde{y})}{p_Y(y)}$$

$$- \beta_2 p_X(x) \sum_{\tilde{y}, y} p_{\tilde{Y}}(\tilde{y}) p_{K_2}(y \mid \tilde{y}) \frac{p_{K_* \circ K_2}(y \mid \tilde{x})}{p_Y(y)}$$

$$= \beta_2 p_X(x) \sum_{\tilde{y}, y} p_{K_* \circ K_2}(\tilde{y}, y \mid \tilde{X} = \tilde{x}) \log \frac{p_{K_2}(y \mid \tilde{y})}{p_Y(y)} - \beta_2 p_X(x) \tag{20}$$

$$\frac{\partial A_4}{p_{K_1}(\tilde{x} \mid x)} = \lambda_1(x) \tag{21}$$

$$\frac{\partial A_5}{p_{K_1}(\tilde{x} \mid x)} = 0 \tag{22}$$

*Taking equations (18)-(22) into (13), we obtain the first-order equation*

$$\frac{\partial L}{\partial p_{K_1}(\tilde{x} \mid x)} = p_X(x) \Bigg\{ -\mathbb{E}_{Y \mid X = x \sim p_{K_*}} \frac{p_{K_* \circ K_2}(Y \mid \tilde{X} = \tilde{x})}{p_K(Y \mid X = x)} + \beta_1 \log \frac{p_{K_1}(\tilde{x} \mid x)}{p_{\tilde{X}}(\tilde{x})}$$

$$+ \beta_2 \mathbb{E}_{\tilde{Y}, Y \mid \tilde{X} = \tilde{x}} \log \frac{p_{K_2}(Y \mid \tilde{Y})}{p_Y(Y)} + \tilde{\lambda}_1(x) \Bigg\} = 0, \tag{23}$$

*where $\tilde{\lambda}(x) = \lambda_1(x) / p_X(x) - \beta_2$. Rearranging the terms in Equation (23), we obtain*

$$\log \frac{p_{K_1}(\tilde{x} \mid x)}{p_{\tilde{X}}(\tilde{x})} = \frac{1}{\beta_1} \Bigg\{ -\tilde{\lambda}_1(x) + \mathbb{E}_{Y \mid X = x \sim p_{K_*}} \frac{p_{K_* \circ K_2}(Y \mid \tilde{X} = \tilde{x})}{p_K(Y \mid X = x)} - \beta_2 \mathbb{E}_{\tilde{Y}, Y \mid \tilde{X} = \tilde{x}} \log \frac{p_{K_2}(Y \mid \tilde{Y})}{p_Y(Y)} \Bigg\}$$

*which implies Equation (2).*

*Similarly, taking derivatives with respect to $p_{K_2}(y \mid \tilde{y})$ for given $\tilde{y}$ and $y$, it can be verified that*

$$\frac{\partial p_K(y \mid x)}{\partial p_{K_2}(y \mid \tilde{y})} = p_{K_1 \circ K_*}(\tilde{y} \mid x)$$

$$\frac{\partial L}{\partial p_{K_2}(y \mid \tilde{y})} = -\sum_x p_X(x) p_{K_*}(y \mid x) \frac{p_{K_1 \circ K_*}(\tilde{y} \mid x)}{p_K(y \mid x)} + \beta_2 p_{\tilde{Y}}(\tilde{y}) \log \frac{p_{K_2}(y \mid \tilde{y})}{p_Y(y)} + \lambda_2(\tilde{y})$$

$$= -\mathbb{E}_{X \sim p_X} \frac{p_{K_*}(y \mid X) \cdot p_{K_1 \circ K_*}(\tilde{y} \mid X)}{p_K(y \mid X)} + \beta_2 p_{\tilde{Y}}(\tilde{y}) \log \frac{p_{K_2}(y \mid \tilde{y})}{p_Y(y)} + \lambda_2(\tilde{y}). \tag{24}$$

*Letting Equation (24) be zero and rearranging it, we obtain Equation (3).*

**Proof 2 (Proof of Theorem 2)** *We define the following functional of four variables:* $p_{K_1}, p_{K_2}, h_1, h_2,$

$$J(p_{K_1}, p_{K_2}, h_1, h_2) = -\sum_{x,y} p_X(x) p_{K_*}(y \mid x) \log p_K(y \mid x) \tag{25}$$

$$+ \beta_1 \sum_{x,\tilde{x}} p_X(x) p_{K_1}(\tilde{x} \mid x) \log \frac{p_{K_1}(\tilde{x} \mid x)}{h_1(\tilde{x})}$$

$$+ \beta_2 \sum_{\tilde{y},y} p_{\tilde{Y}}(\tilde{y}) p_{K_2}(y \mid \tilde{y}) \log \frac{p_{K_2}(y \mid \tilde{y})}{h_2(y)}. \tag{26}$$

*We will use the following known result (Cover, 1999, Lemma 10.8.1). Suppose that $X$ and $Y$ have a joint distribution with density $p_{XY}$, and the marginal densities are $p_X, p_Y$, respectively. Then a density function $r_Y$ of $y$ that minimizes the KL-divergence $D(p_{XY}, p_X r_Y)$ is the marginal distribution $p_Y$. This result implies that minimizing the objective function in (1) can be written as a quadruple minimization*

$$\min_{p_{K_1}, p_{K_2}, h_1, h_2} J(p_{K_1}, p_{K_2}, h_1, h_2). \tag{27}$$

*It can be verified from (23) and its preceding identities that*

$$\frac{\partial^2 J}{\partial p_{K_1}(\tilde{x} \mid x)^2} = p_X(x) \mathbb{E}_{Y|X=x \sim p_{K_*}} \frac{p_{K_* \circ K_2}(Y \mid \tilde{x})}{p_K(Y \mid x)^2} \frac{\partial p_K(Y \mid x)}{\partial p_{K_1}(\tilde{x} \mid x)} + \beta_1 \frac{p_X(x)}{p_{K_1}(\tilde{x} \mid x)}$$

$$= p_X(x) \mathbb{E}_{Y|X=x \sim p_{K_*}} \frac{p_{K_* \circ K_2}(Y \mid \tilde{x})^2}{p_K(Y \mid x)^2} + \beta_1 \frac{p_X(x)}{p_{K_1}(\tilde{x} \mid x)} \tag{28}$$

$$\frac{\partial^2 J}{\partial p_{K_2}(y \mid \tilde{y})^2} = \mathbb{E}_{X \sim p_X} \frac{p_{K_*}(y \mid X) \cdot p_{K_1 \circ K_*}(\tilde{y} \mid X)}{p_K(y \mid X)^2} \frac{\partial p_K(y \mid X)}{\partial p_{K_2}(y \mid \tilde{y})} + \beta_2 \frac{p_{\tilde{Y}}(\tilde{y})}{p_{K_2}(y \mid \tilde{y})} \tag{29}$$

$$= \mathbb{E}_{X \sim p_X} \frac{p_{K_*}(y \mid X) \cdot p_{K_1 \circ K_*}(\tilde{y} \mid X)^2}{p_K(y \mid X)^2} + \beta_2 \frac{p_{\tilde{Y}}(\tilde{y})}{p_{K_2}(y \mid \tilde{y})} \tag{30}$$

$$\frac{\partial^2 J}{\partial h_1(\tilde{x})^2} = \beta_1 \sum_x \frac{p_X(x) p_{K_1}(\tilde{x} \mid x)}{h_1(\tilde{x})^2} \tag{31}$$

$$\frac{\partial^2 J}{\partial h_2(y)^2} = \beta_2 \sum_{\tilde{y}} \frac{p_{\tilde{Y}}(\tilde{y}) p_{K_2}(y \mid \tilde{y})}{h_2(y)^2} \tag{32}$$

*Thus, $J(p_{K_1}, p_{K_2}, h_1, h_2)$ is convex in each of the variables.*

*We begin with a choice of initial $p_{K_2}, h_1, h_2$, and calculate the $p_{K_1}$ that minimizes the objective. Using the method of Lagrange multipliers for this minimization (in a way similar to (13)), we obtain the solution of $p_{K_1}$ shown in the first equation of Line 3, Algorithm 1. Similarly, we obtain the second equation in Algorithm 1. For the conditional distributions $p_{K_1}$ and $p_{K_2}$, we then calculate the marginal distributions $h_1$ (of $\tilde{x}$) that minimizes (26). Note that the terms of (26) involving $h_1$ may be rewritten as*

$$\beta_1 \sum_{x,\tilde{x}} p(x,\tilde{x}) \log \frac{p(x,\tilde{x})}{p(x) h_1(\tilde{x})}$$

*which, by the aforementioned lemma, is minimized by the third equation of Line 3, Algorithm 1. Similar arguments apply for $h_2$. Consequently, each iteration step in Algorithm 1 reduces $J$. By the non-negativeness of KL-divergence, $J + c \geq L \geq 0$, where $L$ is in (1) and $c$ is introduced in (13). Therefore, $J$ has a lower bound, and the algorithm will converge to a minimum. Note that $J(p_{K_1}, p_{K_2}, h_1, h_2)$ is convex in each of the variables independently but not in the variables' product space. The current proof does not imply the convergence to a global minimum.*

**Proof 3 (Proof of Theorem 3)** *Similar to the technique used in the above proof of Theorem 2, we cast the optimization problem in (6) as a double minimization with respect to $(p_{K_2}, h_2)$,*

$$J(p_{K_2}, h_2) \triangleq -\sum_{x,y} p_X(x) p_{K_*}(y \mid x) \log p_K(y \mid x) + \beta_2 \sum_{\tilde{y},y} p_{\tilde{Y}}(\tilde{y}) p_{K_2}(y \mid \tilde{y}) \log \frac{p_{K_2}(y \mid \tilde{y})}{h_2(y)}.$$

*We only need to check that $J$ is strongly convex in its arguments. Direct calculations show that*

$$\frac{\partial^2 J}{\partial p_{K_2}(y \mid \tilde{y})^2} = \sum_x p_X(x) p_{K_*}(y \mid x) \frac{p_{K_*}^2(\tilde{y} \mid x)}{p_K^2(y \mid x)} + \beta_2 \frac{p_{\tilde{Y}}(\tilde{y})}{p_{K_2}(y \mid \tilde{y})}$$

$$= \sum_{x: f(x) = y, y = \tilde{y}} p_X(x) \frac{1}{p_K^2(y \mid x)} + \beta_2 \frac{p_{\tilde{Y}}(\tilde{y})}{p_{K_2}(y \mid \tilde{y})}$$

$$= \frac{p_{\tilde{Y}}(\tilde{y})}{p_{K_2}^2(y \mid \tilde{y})} \mathbb{1}_{y = \tilde{y}} + \beta_2 \frac{p_{\tilde{Y}}(\tilde{y})}{p_{K_2}(y \mid \tilde{y})}$$

$$\frac{\partial^2 J}{\partial h_2(y)^2} = \beta_2 \frac{p_Y(y)}{h_2(y)^2}$$

$$\frac{\partial^2 L}{\partial p_{K_2}(y \mid \tilde{y}) \partial h_2(y)} = -\beta_2 \frac{p_{\tilde{Y}}(\tilde{y})}{h_2(y)}.$$

*The above equations indicate that the determinant of the Hessian satisfies*

$$\frac{\partial^2 J}{\partial p_{K_2}(y \mid \tilde{y})^2} \cdot \frac{\partial^2 J}{\partial h_2(y)^2} - \left\{ \frac{\partial^2 L}{\partial p_{K_2}(y \mid \tilde{y}) \partial h_2(y)} \right\}^2$$

$$= \beta_2 \frac{p_{\tilde{Y}}(\tilde{y}) p_Y(y)}{p_{K_2}(y \mid \tilde{y}) h_2(y)^2} \mathbb{1}_{y = \tilde{y}} + \beta_2^2 \frac{p_{\tilde{Y}}(\tilde{y}) p_Y(y)}{p_{K_2}(y \mid \tilde{y}) h_2(y)^2} \left\{ 1 - \frac{p_{\tilde{Y}, Y}(\tilde{y}, y)}{p_Y(y)} \right\},$$

*which further implies the convexity of $J$ in the product space of $p_{K_2}$ and $h_2$.*

### A.4  VISUALIZATION OF ALGORITHM 2

We provide a toy example to visualize Algorithm 2. In the simulation, we choose an alphabet of size 100, and $p_{\tilde{Y}}$ as described by $r \in [0, 1]^a$ is uniform-randomly generated from the probability simplex. We independently replicate the experiment 50 times, each time running Algorithm 2 for 30 iterations, and calculate the average of the following results. First, we record $\|\boldsymbol{P}^{(t+1)} - \boldsymbol{P}^{(t)}\|_1 / a$ at each iteration $t$, which traces the convergence of the estimated transition probabilities. Second, we record the final transition probability matrix into a heat-map where $P_{y, \tilde{y}}$ means the estimated $p_{K_2}(y \mid \tilde{y})$. The experiments are performed for $\beta = 100, 10, 1$, corresponding to columns 1-3. The plots indicate the convergence of the algorithm, though the rate of convergence depends on $\beta$. They also imply the expected result that a small $\beta$ induces an identity transition while a large $\beta$ induces $\tilde{Y}$ that is nearly independent with $Y$.

### A.5  DATA STUDY: NEWS TEXT CLASSIFICATION

In this experimental study, we use the '20-newsgroups' dataset provided by *scikit-learn* open-source library (Scikit-learn, 2020d), which comprises news texts on various topics. The experiment is intended to illustrate the utility-privacy tradeoff and the optimality of our proposed solution compared with other methods. For better visualization we pick up the first four topics (in alphabetic order), which are 'alt.atheism', 'comp.graphics', 'comp.os.ms-windows.misc', 'comp.sys.ibm.pc.hardware'. Suppose that the service Alice provides is to perform text-based clustering, which takes text data as input and returns one of the four categories (denoted by $0, 1, 2, 3$) as output. The texts are transformed into vectors of numerical values using the technique of term frequency-inverse document frequency (TF-IDF) (Rajaraman & Ullman, 2011). In the transformation, metadata such as headers, signature blocks, and quotation blocks are removed. To evaluate the out-sample utility, we split the data into two parts using the default option provided in (Scikit-learn, 2020d), which results in a training part (2245 samples, 49914 features) and a testing part (1494 samples, 49914 features). The above split between the training and testing is based upon messages posted before and after a specific date.

Alice trains a classifier using the Naive Bayes method and records the frequency of observing each category $[0.22\,0.27\,0.21\,0.30]$ ($r$ in Algorithm 2). Then, Alice runs the OIL-Y Algorithm (under a given $\beta_2$) to obtain the transition probability matrix $P \in [0, 1]^{4 \times 4}$. In other words, the effective system provided by Alice is the cascade of the learned classifier, and $P$ determines the Markov

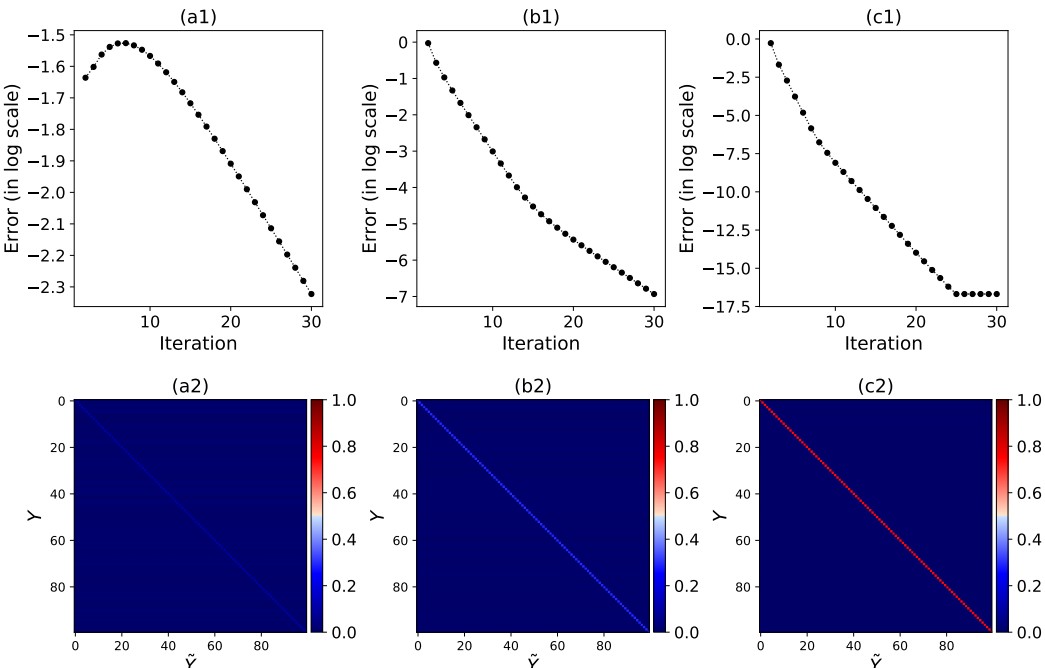

Figure 3: Visualization of Algorithm 2 in terms of the convergence (row 1) and the final transition probabilities (row 2), for $\beta = 100, 10, 1$ (corresponding to three columns).

transition. Alice's resulting out-sample performance from the testing data is recorded in Figure 4a, where we considered different $\beta$'s summarized in Table 1. As we expected, a larger value of $\beta_2$ cuts off more information propagated from $\tilde{Y}$ to $Y$, resulting in a degraded out-sample performance of Alice's effective system.

We also visualize the model privacy-utility tradeoff by the following procedure. First, we approximate the utility that quantifies the useful information conveyed by Alice. With Alice's trained model and the optimally laundered $Y$ (from training data), we retrain another Naive Bayes classifier and generate predictions on the testing data, denoted by $y_K^{\mathrm{pred}}$. Meanwhile, we apply Alice's authentic model to generate predictions on the testing data, denoted by $y_{K_*}^{\mathrm{pred}}$. We approximate the model utility as the accuracy measure between $y_K^{\mathrm{pred}}$ and $y_{K_*}^{\mathrm{pred}}$. The model utility can be approximated by other measures. We also considered retraining methods such as tree-based classifiers and average F1-score in computing the model utility, and the results are consistent in the data experiments. Second, we approximate the privacy leakage as Alice's prediction accuracy on the testing data. Intuitively speaking, for a given utility, larger out-sample prediction accuracy indicates less information laundered, indicating a higher privacy leakage of Alice's internal model. We plot the model leakage against utility obtained from our proposed solution in Figure 4b.

For comparison, we considered a benchmark method described below. The conditional probability mass function $p_{K_2}(\cdot \mid \tilde{y})$ given each $\tilde{y}$ is independently drawn from a Dirichlet distribution with parameters $[b, \ldots, b, a, b, \ldots, b]$, where $a$ is the $\tilde{y}$th entry. An interpretation of the parameter is that a larger $a/b$ favors a larger probability mass at $y = \tilde{y}$ (and thus less noise). We consider different pairs of $(a, b)$ so that the tradeoff curve matches the counterpart curve from our proposed method. The curve is averaged over 50 independent replications. As shown in Figure 4b, the results indicate that our proposed solution produces less leakage (and thus better privacy) for a given utility.

We also plot heatmaps illustrating the transition laws $p_{K_2}(y \mid \tilde{y})$ obtained from the proposed information laundering in Figure 5. We considered two cases, where there are 20% class-0 labels, and where there are 1% class-0 labels (by removing related samples from the original dataset). Intuitively, once we reduce the size of class-0 data in (b), the transition probabilities $p_{K_2}(0 \mid \tilde{y})$ for each $\tilde{y}$ should be smaller compared with those in (a) as class-0 is no longer 'important'. Our expectation

Table 1: Summary of the tradeoff parameters used for the OIL-Y algorithm and random benchmark from Dirichlet distributions (averaged over 50 independent replications), and the corresponding model utility (as evaluated by the closeness of Alice's authentic and effective systems), as well as the model privacy leakage (as evaluated by Alice's out-sample accuracy).

| | $\beta$ | 0 | 1 | 2 | 5 | 20 | 50 |
|---|---|---|---|---|---|---|---|
| Proposed | Utility | 1.00 | 0.86 | 0.78 | 0.68 | 0.46 | 0.30 |
| | Leakage | 0.79 | 0.64 | 0.53 | 0.45 | 0.35 | 0.30 |
| Random Benchmark | $a, b$ | 100, 1 | 20, 1 | 10, 1 | 5, 2 | 5, 3 | 10, 10 |
| | Utility | 0.96 | 0.88 | 0.79 | 0.49 | 0.39 | 0.23 |
| | Leakage | 0.77 | 0.70 | 0.62 | 0.40 | 0.34 | 0.27 |

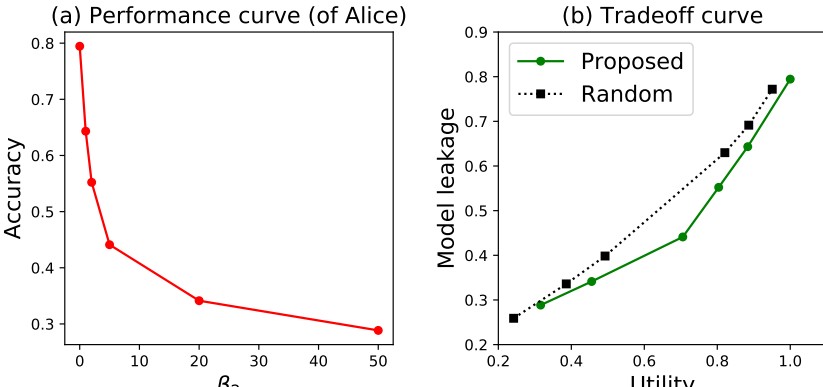

Figure 4: Visualization of (a) Alice's out-sample performance against the tradeoff parameter $\beta_2$ in Information Laundering, and (b) Alice's model utility-privacy tradeoffs under the information laundering technique and the random benchmark using Dirichlet-generated transition laws. Detailed parameters are summarized in Table 1.

is aligned with Figure 5, where the first row in (b) are indicated by darker colors compared with that in (a), meaning that the class-0 is less likely to be observed.

## A.6 DATA STUDY: LIFE EXPECTANCY REGRESSION

In this experimental study, we use the 'life expectancy' dataset provided by *kaggle* open-source data (Kaggle, 2020), originally collected from the World Health Organization (WHO). The data was collected from 193 countries from 2000 to 2015, and Alice's model is a linear regression that predicts life expectancy using potential factors such as demographic variables, immunization factors, and mortality rates. This experiment is intended to illustrate the utility-privacy tradeoff and our proposed solution in regression contexts.

In the regression model, we quantize the output alphabet $\mathcal{Y}$ by 30 points equally-spaced in between $\mu \pm 3\sigma$, where $\mu, \sigma$ represent the mean and the standard deviation of $Y$ in the training data. We then applied a similar procedure as in Subsection A.6, except that we use the empirical $R^2$ score as the underlying measure of utility and leakage. The empirical $R^2$ score has been commonly used for evaluating regression performance, and it can be negative, meaning that the predictive performance is worse than sample mean-based prediction (Scikit-learn, 2020a). In particular, we obtain tradeoff curves in Figure 6, where we compared the information laundering results based on the proposed technique and Dirichlet-based technique (similar to that in Subsection A.6). The different $\beta$'s and Dirichlet parameters are summarized in Table 2. The detailed performance values are also summarized in Table 2.

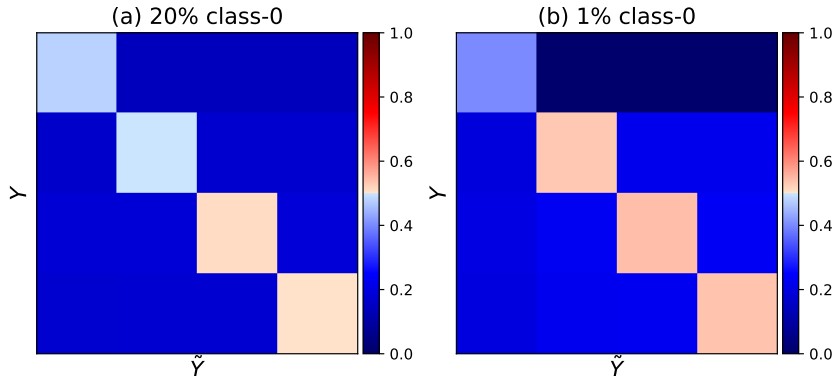

Figure 5: Heatmap showing the transition law $p_{K_2}(y \mid \tilde{y})$ for information laundering, under (a) 20% of class-0 labels, and (b) 1% of class-0 labels. In contrast with the case (a), the class-0 is negligible in (b) and thus the transition probabilities $p_{K_2}(0 \mid \tilde{y})$ for each $\tilde{y}$ becomes smaller (as indicated by darker colors).

Table 2: Summary of the tradeoff parameters used for the OIL-Y algorithm and random benchmark from Dirichlet distributions (averaged over 50 independent replications), and the corresponding model utility (as evaluated by the closeness of Alice's authentic and effective systems), as well as the model privacy leakage (as evaluated by Alice's out-sample accuracy). The underlying metric used is the empirical $R^2$, which can be less than zero.

|  | $\beta$ | 0 | 1 | 2 | 5 | 8 | 20 |
|---|---|---|---|---|---|---|---|
| Proposed | Utility | 0.99 | 0.92 | 0.84 | 0.62 | 0.48 | 0.35 |
|  | Leakage | 0.79 | 0.42 | 0.09 | $-0.26$ | $-0.45$ | $-0.51$ |
| Random Benchmark | $(a, b)$ | $10000, 1$ | $200, 5$ | $100, 5$ | $100, 8$ | $100, 10$ | $100, 20$ |
|  | Utility | 0.99 | 0.77 | 0.58 | 0.42 | 0.36 | 0.15 |
|  | Leakage | 0.78 | 0.10 | $-0.07$ | $-0.15$ | $-0.17$ | $-0.22$ |

To illustrate the impact of tradeoffs, we considered two cases corresponding to $\beta_2 = 1$ and $\beta_2 = 20$. We compute the transition laws $p_{K_2}(y \mid \tilde{y})$ obtained from Algorithm 2 and illustrate them in the first row of Figure 5. We also take the snapshot at the year $\tilde{Y} = 69$ and plot the conditional density function $p_{K_2}(\cdot \mid \tilde{Y} = 69)$ (as approximated by the quantizers) in the second row of Figure 5. The visualized results are aligned with our expectation that a larger penalty of model leakage will cause a more dispersed transition law.

## A.7 DATA STUDY: MITIGATION OF ADVERSARIAL ATTACKS

In this experimental study, we demonstrate the use of information laundering in mitigating two model extraction attacks studied in (Tramèr et al., 2016). We consider settings where the target model is a classifier. The first attack is the Retraining attack. The adversary Bob sends random queries and receives class labels to train a local model. Bob does not need to know Alice's model architecture. The second attack is the equation-solving attack. Alice's model is assumed to be a Logistic classifier, and Bob knows about it. Bob sends random queries in this attack and receives class probabilities (a vector on a simplex). Bob then solves a linear equation to obtain the coefficients of Alice's Logistic model. We note that there exist more types of attacks suitable for various specific settings (see., e.g., Tramèr et al., 2016; Juuti et al., 2019).

The experiments in this section indicate the following two points. First, an adversary needs more samples to achieve the same utility under the information laundering (Figure 8). Second, the effects of attack and laundering depend on the particular models specified by the adversary and target (Figure 9).

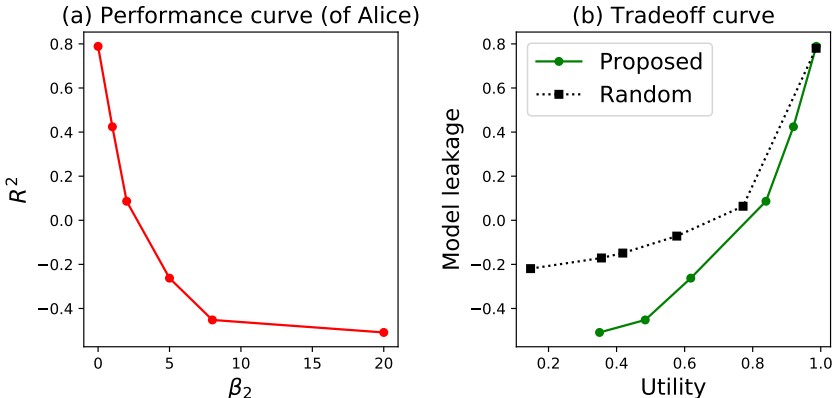

Figure 6: Visualization of (a) Alice's out-sample performance against the tradeoff parameter $\beta_2$ in Information Laundering, and (b) Alice's model utility-privacy tradeoffs under the information laundering technique and the random benchmark using Dirichlet-generated transition laws. Detailed parameters are summarized in Table 2.

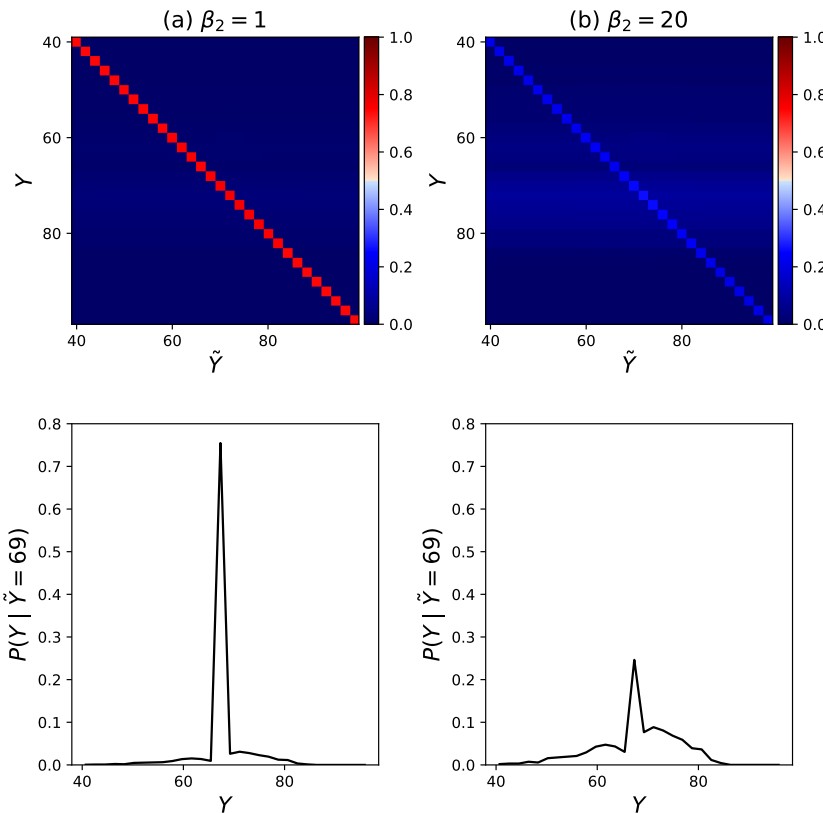

Figure 7: Heatmap (row 1) showing the transition laws optimized from information laundering, under (a) $\beta_2 = 1$, and (b) $\beta_2 = 20$. The snapshots of probability mass functions of $Y$ conditional on $\tilde{Y} = 69$ are also visualized (row 2).

The experimental details are given below. In Figure 8(a), Alice uses half of the Breast Cancer dataset (Scikit-learn, 2020b) (standardized) to train a Logistic classification model. Bob queries the class labels with standard Gaussian random input and locally trains another Logistic classifier. In responding to Bob, Alice employs different levels of laundering. The utility is defined as the

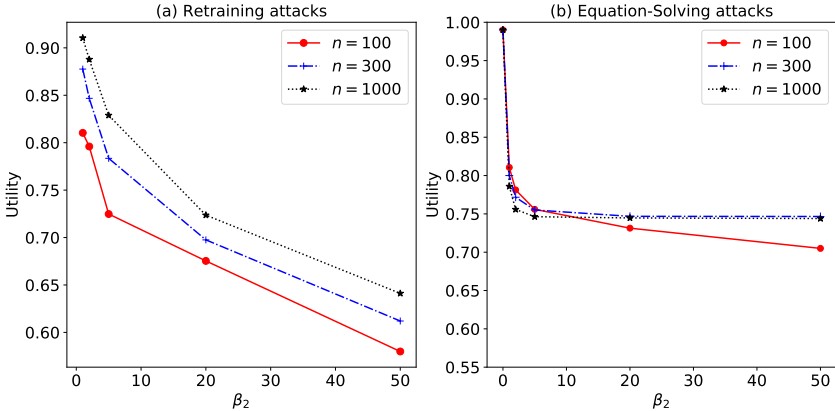

Figure 8: An adversary's utility against the laundering parameter $\beta_2$ in the contexts of (a) Retraining attack with random queries, and (b) Equation-Solving attack. In each plot, the three curves represent query sizes $n = 100, 300, 1000$, respectively.

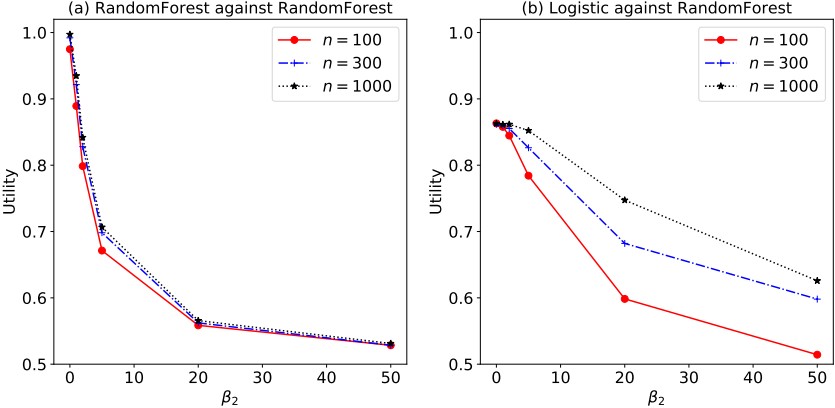

Figure 9: An adversary's utility against the laundering parameter $\beta_2$ in the contexts of Retraining attacks, where (a) both the adversary and the target use the Random Forest model, and (b) the adversary uses the Logistic model while the target uses the Random Forest. In each plot, the three curves represent query sizes $n = 100, 300, 1000$, respectively.

percentage of agreement of Alice's and Bob's models tested on the other half of the dataset. In Figure 8(b), Bob sends the same random queries and solves a linear equation to estimate Alice's Logistic coefficients. From the results, we can see that Bob's performance is better than that in Figure 8(a). It is mainly due to the substantial side information and the least-squares estimate of Bob.

In Figure 9, Alice used half of the simulated Moons dataset (Scikit-learn, 2020c) (with 1000 samples, 0.1 standard deviation for the noise) to train a Random Forest model. The other half of the data are reserved to evaluate Bob's utility. Suppose that Bob uses the above Retraining attack, but with two different models. In Figure 9(a), Bob uses the Random Forest classifier, which has the expressive power to extract Alice's model. In Figure 9(b), Bob uses the Logistic classifier, which has a linear decision boundary. Bob's model in (b) is inadequate because the Moons data is not linearly separable. Figure 9 shows that the influence of information laundering on the adversarial query size needed to maintain the same utility largely depends on Bob's model choices. Also, Bob's inadequate model may show more robustness against laundered information even though it does not perform satisfactorily on cleaned data.

