# OpenReview forum: "Information Laundering for Model Privacy"
_ICLR.cc/2021/Conference — ICLR 2021 Spotlight_

### Official Review · AnonReviewer2 · 2020-10-28
**This paper provides a new framework for model privacy**

**Rating:** 7
**Confidence:** 2

**Review:**

Summary: This paper studies model privacy and its privacy and utility tradeoff. In particular, the authors proposed information laundered model where the input and output of the true model are perturbed.  The objective is to minimize the KL divergence between the true model and laundered model with mutual information between input and output as constraints. Theoretically, they show the optimal condition of the above optimization problem and provides an iterative algorithm.


Pros: 1. Model privacy is indeed an interesting problem. This paper is well written.


Questions: 1. The authors claim the proposed framework and algorithm can be also applied to continuous space. However, it is not clear to me how algorithm 1 would work in continuous space.

---

> ### Author Response · Authors · 2020-11-18
> **Detailed responses to each comment of Reviewer2**
>
> Thank you for the constructive comments.
>
> [Comment on the continuous space]
> We have checked that the same theoretical proofs apply to the continuous alphabets (with summation replaced with integral). The theorems and Algorithm 1 would remain valid for the continuous case.
> Without any computational concern, Algorithm 1 can be implemented by numerical integration (e.g., to use Monte Carlo techniques).  Nevertheless, the discrete case is more suitable for practical operations, e.g., Algorithm 2 written in a matrix form. So we would suggest discretizing continuous alphabets to implement the information laundering algorithm. We did so in the regression-based experimental study (Section A.6).

---

### Official Review · AnonReviewer1 · 2020-10-28
**Interesting idea, not well-situated in the existing literature**

**Rating:** 6
**Confidence:** 4

**Review:**

Paper summary:
This paper aims to tackle the problem of model stealing/extraction, as in stealing a model that is deployed remotely, through API access. The threat model that they are aiming to protect against is not well-defined. The proposed method is information theoretic. They propose adding two modules (kernels) before and after the main deployed model, to "launder" information.  The loss function for achieving the desired modules consists of two main terms, for utility, and privacy of the model. The utility term tries to keep the expected value of the output being accurate high, while the privacy term (which are actually two terms for the two modules) try to decrease the MI between the true output/input and the laundered ones. They then offer an iterative approach for minimizing the loss.

pros:
+ I like how the approach is model-agnostic, you don't seem to need to change the model, you just add some kernels. And it can be applied to any model.

+ The suggested approach is intuitive and sound.

cons:
- The paper is not well-situated, given prior work. I am not completely familiar with the model stealing literature either, but with a search I found numerous papers since 2016 that either propose model stealing attacks, or mitigations. The paper does not qualitatively nor quantitatively compare their approach to any other prior protection approaches, such as [1], [2] or [3].

- The paper does not conduct any ablation studies or experiments into the way their approach affects the model accuracy, and how effective it is to existing attacks, such as the Tramer et al. attack. This makes at really hard to evaluate the claims made about privacy and utility, in action.

- Mutual Information and Expected Value (used in the loss function) are both average notion. I wonder what this means for the provided protection. Would the model perform really badly on some few examples? would there be some sample inputs that could extract a big part of the model? I think a study of this, or at least a discussion would be very helpful.

One final note, it would be nice to have the threat model well-defined, it would really help better communicate what the protection is supposed to do, and what we should be expecting/not expecting of it.

References
[1] Juuti M, Szyller S, Marchal S, Asokan N. PRADA: protecting against DNN model stealing attacks. In2019 IEEE European Symposium on Security and Privacy (EuroS&P) 2019 Jun 17 (pp. 512-527). IEEE.

[2] Orekondy T, Schiele B, Fritz M. Prediction Poisoning: Towards Defenses Against DNN Model Stealing Attacks. InInternational Conference on Learning Representations 2019 Sep 25.

[3] Lee T, Edwards B, Molloy I, Su D. Defending against machine learning model stealing attacks using deceptive perturbations. arXiv preprint arXiv:1806.00054. 2018 May 31.


--------- Comments after reading the author response:
I thank the authors for adding the experiments and applying the suggested modifications! I have updated my score based on the changes and the clarifications made on the related work, and also the results of the mounted attacks.

---

> ### Author Response · Authors · 2020-11-18
> **Detailed responses to each comment of Reviewer1**
>
> Thank you for the constructive comments.
>
> [Comment on the related work]
> As you suggest, we included the following related work in the revision. These papers all contain very interesting ideas. We qualitatively compare each of them with information laundering.
>
> [1] "Model extraction warning in mlaas paradigm" 2018
>
> Comparison: This work developed a heuristic warning-based method to alleviate model extraction attacks. The service provider continuously monitors the information gain and raises the alarm when the monitored statistics are unusual. Thus, compared with information laundering, the mitigation of model leakage in [1] is by cutting off services for potential adversaries. The tradeoff would be between the false-alarm rate and detection power.
>
> [2] "PRADA: protecting against DNN model stealing attacks" 2019
>
> Comparison: This works developed another warning-based method. The main difference is the statistic used to detect adversaries. More specifically, the work experimentally found that under the null hypothesis (that the user is benign), the pairwise distances between queried data approximately follow the Gaussian distribution. The empirical distribution is no longer Gaussian under some common attacks studied in [2]. Thus, the detection problem is turned into a normality test.
>
> To experimentally compare [1,2] and information laundering, we postulate the following experimental plan (future work).
> There will be tradeoffs between the probability of falsely rejecting the null hypothesis and correctly rejecting the null for any warning-based method. For any fixed decision rule, we perform the attack-defense in several independent replications. To calculate the overall utility, we evaluate the utility of the model trained from all the benign-queried data responded until an alarm is raised (upper limited by a large number), and average it over replications. To calculate the overall leakage, we evaluate the utility of the model trained from all the adversarially queried data responded until an alarm is raised, and then average it over replications. We then compare the privacy-utility tradeoff with information laundering.
>
> [3] "Defending against machine learning model stealing attacks using deceptive perturbations" 2018
>
> Comparison: This work developed a defense strategy for the setting where the target is a classifier, and the adversary queries the probability of each class. The probabilities are maximally perturbed under the constraint that the argmax (namely the most-likely class label) remains the same. The underlying assumption is that the probabilities are not considered as a utility. Compared with [3] that focuses on classification problems, information laundering applies to more general APIs.
>
> [4] "Prediction Poisoning: Towards Defenses Against DNN Model Stealing Attacks" 2019
>
> Comparison: This work also studied the setting where a potential adversary queries probabilities from a classifier. Different from [3], the idea in [4] is to perturb the probabilities (on a simplex) to the direction that does the most harm to the adversary's gradient during its retraining. The idea of perturbing the output is similarly considered in information laundering. A difference is that the constraint of perturbations in [4] used the L2 distance instead of mutual information. Another difference is that the perturbation of the output in [4] is deterministic (given the output) instead of probabilistic in our case. Also, the perturbation strategy depends on the assumption that the target model is a classifier outputting a probability simplex.
>
> Overall, the unique features of information laundering compared with the existing work include the following. First, information laundering is (to the best of our knowledge) the first theoretically-founded framework for enhancing model privacy. Second, the information laundering principle applies to a general target model (not only for classification models), and it is easy to implement due to the plug-and-play nature. As future work, we think that the information laundering could be better customized for specific target models, e.g., by conditional on various side information.
>
> (responses continued in the next box)

---

> > ### Author Response · Authors · 2020-11-18
> > **Detailed responses to each comment of Reviewer1 (Continued)**
> >
> > [Comment on the ablation studies]
> > According to your suggestion, we have included additional experiments to illustrate how information laundering provides mitigation against model extraction attacks (in Section A.7, pages 19-21 in the revised paper).
> > We studied two existing attacks: the retraining attack based on random queries and the equation-solving attack (both from the work of Tramer et al.). The added experiments show that 1) information laundering is effective, in the sense that it requires the adversaries to use more samples to achieve the same utility, and 2) the effects of attack and laundering may depend on how the models are specified (especially in black-box attack scenarios). Detailed discussions are included in the experimental section.
> > We will compare with more types of attacks in future work.
> >
> >
> > [Comment on the vulnerable scenarios]
> > The raised questions are very interesting. We think it is possible to have scenarios where few queries steal a big part of the model. We provide the following example. Suppose that the model is a supervised learning model. It is known (at least to the adversary) that the label at the extreme-value region, e.g., where \norm{x} > 100, is a constant. The adversary may then strategically send the same x that falls into that region and use the queried responses to identify that constant label. Because the generic information laundering concerns the average scenario, unless the side information is conditioned on, the above vulnerable scenario can occur in practice. We have included a related discussion in the revision (Section 5).
> >
> >
> > [Comment on the threat model]
> > We have clarified the threat model in the revised paper (in Section 2).
> > Specifically, the adversary can be a user that can access the target model's API, providing an arbitrary input, and obtaining the output. The output value can be a class label, a regression response, a probability simplex, etc.
> > The adversary aims to use as few queries as possible to construct a model that closely matches the target model. We formalize closeness using the KL divergence. The protection operated by information laundering will increase the adversary's cost in achieving the same utility, and it should work for any attacking model. On the other hand, information laundering does not take into account adversary's information. So it may not work well in the presence of strong side information (as seen in the above example).

---

> > > ### Comment · AnonReviewer1 · 2020-11-24
> > > **Post-rebuttal**
> > >
> > > Thank you very much for the response and the updates. I have updated my score.

---

### Official Review · AnonReviewer5 · 2020-11-09
**A novel approach to a problem of growing interest.  Empirical comparisons to related works would improve the submission.**

**Rating:** 7
**Confidence:** 4

**Review:**

Theoretically:
The paper is both conceptually and formally descriptive.  Combined with appendix details, the authors present a very full picture of their work, which is very much appreciated.  To the best of my knowledge, this appears to be a novel approach to the challenge of adversarial model reconstruction.


Experimentally:
Some experiments reside in the appendices that confirm utility of the theoretical basis of information laundering, however, I believe further experimentation would strengthen the paper significantly.

There appear to be multiple works (Tramer / Papernot / etc.) that provide different strategies for reconstructing private learning models.  A high-value experiment would be to test those methods on a model protected with the information laundering framework.

Additionally, there appear to be other techniques for safeguarding against model reconstruction efforts (some papers shared below).  A comparison between those and information laundering would help inform design decisions that practitioners/users are making about their machine-learning-as-a-service systems.  I’m left wanting to know how information laundering compares to other model-privacy protective measures.  A clear answer to that question would make the paper very strong, in my opinion.


Related work:
While submission length is always a limiting factor, I believe it may be worth mentioning a few other efforts.

‘Federated machine learning: concept and applications’ ACM 2019
Data privacy can be related to the topic of ‘Federated Learning’.  Mention in related work would provide an extra connection to the machine learning community for a security related problem setting.

There appear to be a few more model extraction techniques in literature than are mentioned in related work:

“Model reconstruction from model explanations” FAT* 2019
Swaps out prediction API for ‘explanation API’

“Membership inference attacks against machine learning models” IEEE 2017.
Under the assumption that the adversary has access to data, infer which samples make up training data, and use this information to reconstruct their own model.


Directly relevant related work:
Philosophically speaking, there are some direct competitors that are not addressed, in the sense that they are methods to safeguard against leakage / maintain model privacy.  I only argue that these should be addressed/discussed by the authors, not that these papers invalidate the authors' novelty claims.

“Model extraction warning in mlaas paradigm” ACM 2018
“PRADA: protecting against DNN model stealing attacks” IEEE 2019

Both papers monitor queries and raise alarms when query distributions become adversarially-suspect.  Functionally, preventing adversarial queries but allowing benign queries solves the same problem as information laundering but without reducing utility for all users.  The question of interest would be whether these alarms can be sounded before meaningful information is recovered from the model.  I suspect information laundering provides a stronger safeguard against adversarially model reconstruction efforts than these warning systems, but experimental evidence would make the case even stronger.

Overall:
The paper reads very well.  Further experimentation would help relate information laundering to other related efforts, however, I believe the paper can stand on its own as more of a theoretical investigation rather than an empirical one.  For that reason, I believe the paper is strong enough for acceptance as-is.

---

> ### Author Response · Authors · 2020-11-18
> **Detailed responses to each comment of Reviewer5**
>
> Thank you for the constructive comments.
>
> [Comment on the theory]
> Thank you for your support.
>
> [Comment on the experiment]
> According to your suggestion, we have included further experiments to demonstrate how information laundering will affect the adversarial model extraction (in Section A.7, pages 19-21 in the revised paper). Specifically, we studied two attacks, namely the retraining attack based on random queries and the equation-solving attack (from the work of Tramer et al.). The added experiments mainly show that 1) adversaries need more samples to achieve the same utility under the information laundering, and 2) the effects of attack and laundering depend on how the adversary and target specify their models (in black-box attack scenarios). We also pointed out some interesting observations from the experimental study.
> We realized that there are many more types of attacks suitable for various settings during our literature studies. We introduced some of them in the updated paper and discussed possible future work.
>
> [Comment on the comparison of information laundering and other model-privacy protective measures]
> In the updated paper, we have included the following work that aims to safeguard against model leakage (in Section 1).
>
> [1] "Model extraction warning in mlaas paradigm" 2018
>
> [2] "PRADA: protecting against DNN model stealing attacks" 2019
>
> [3] "Defending against machine learning model stealing attacks using deceptive perturbations" 2018
>
> [4] "Prediction Poisoning: Towards Defenses Against DNN Model Stealing Attacks" 2019
>
> The above papers are all closely related and contain exciting ideas.
> As pointed out by you, the first two papers use warning-based methods to alleviate model extraction attacks. In particular, [1] developed a warning method based on monitoring either the information gain or user summaries. [2] developed a technique based on detecting the non-Gaussian distribution based on the pairwise distances between queried data. We have not been able to implement their systems for comparison (partly because we did not find the corresponding open-source codes) information. We postulate the following experimental plan for future work.
>
> For any warning-based method, there will be tradeoffs between the false alarm rate (probability of falsely rejecting the null hypothesis) and detection power (probability of correctly rejecting the null). For any fixed decision rule (e.g., the significance level specified for the tests), we replicate the attack-defense in several independent trials. To calculate the overall utility, we evaluate the utility of the model trained from all the benign-queried data responded until an alarm is raised (upper limited by a large number), and average it over replications. To calculate the overall leakage, we evaluate the utility of the model trained from all the adversarially queried data responded until when an alarm is raised and average it over replications. We then compare the privacy-utility tradeoff with information laundering.
>
> The work in [3] developed a defense strategy for the setting where the target is a classifier, and the adversary queries the probability of each class. The probabilities are maximally perturbed under the constraint that the argmax (namely the most-likely class label) remains the same.
> The work in [4] studied a similar setting as in [3] (querying probabilities from a classifier), but from a different perspective. The main idea is to perturb the probabilities (y on a simplex) to the direction that contributes the least to the adversary's optimal gradient during its retraining. The idea of perturbing the output y is similarly considered in the information laundering, though the constraint of perturbations in [4] used the L2 distance while we used the mutual information.
>
> We summarize the unique features of information laundering compared with existing work as follows. 1) The information laundering is (to the best of our knowledge) the first theoretically-founded framework for enhancing model privacy. 2) The information laundering principle applies to a general target model (not only for classification models). 3) The model does not need to assume what model a potential adversary uses and does not need local retraining to launder the information. It only needs a plug-and-play module (K1 and K2 in the paper) for easy implementation. Meanwhile, for point 3, we also think that information laundering could be customized for specific target models, e.g., by conditional on various side information. We think it as an interesting future direction.
>
> [Comment on the related work in a broader privacy community]
> As you suggest, we have included more related work on other types of privacy issues (in Section 1.1), including the following work.
>
> [5] "Federated machine learning: concept and applications" 2019
> [6] "Model reconstruction from model explanations" 2019
> [7] "Membership inference attacks against machine learning models" 2017

---

### Author Response · Authors · 2020-11-18
**Summary of the revisions**

We thank all three reviewers for their encouraging words and constructive comments.
According to the reviews, we have mainly made the following efforts in the recent few days.
First, we studied the mentioned papers on adversarial model extraction and incorporated them in the current paper.
Second, we made some experimental studies by applying the information laundering to some existing attacks. The experiments were by no means comprehensive, so we included additional discussions on possible future work.
Third, we included discussions comparing the information laundering and some existing preventive techniques and highlighted their similarities and differences.
The major revisions are marked with blue color in the revised paper.
Next, we address each comment of three reviewers.

---

### Decision · Program_Chairs · 2021-01-07
**Final Decision**

**Decision:**

Accept (Spotlight)

**Comment:**

This clearly written paper has been constructively evaluated by three expert reviewers who provided at least two very detailed and informative summaries. The authors have addressed the inquiries raised by the reviewers in a comprehensive fashion, and at least one reviewer has updated their score as a result of those detailed rebuttals. In spite of some outstanding limitations, including a somewhat limited view of the relation of the proposed approach to existing alternatives, the reviewers are consistent in suggesting that this work is sufficiently mature to be considered for the inclusion in the program of ICLR 2021. I concur with that and recommend accepting this paper.